# Effects of Probiotic Short-Term Regiment on Oral Health Parameters in Children: A Pilot Randomized Controlled Trial

**DOI:** 10.3390/nu17223604

**Published:** 2025-11-18

**Authors:** Edouard Starck, Vanessa Machado, João Botelho, Luís Proença, Helena Barroso, Carla Ascenso, Cecília Rozan

**Affiliations:** Egas Moniz Center for Interdisciplinary Research (CiiEM), Egas Moniz School of Health & Science, 2829-511 Almada, Portugal; edouard.starck974@gmail.com (E.S.); vmachado@egasmoniz.edu.pt (V.M.); jbotelho@egasmoniz.edu.pt (J.B.); lproenca@egasmoniz.edu.pt (L.P.); mhbarroso@egasmoniz.edu.pt (H.B.); carla.ascenso@egasmoniz.edu.pt (C.A.)

**Keywords:** children, dental caries, probiotics, *Streptococcus mutans*, oral microbiota, randomized controlled trial, *Lactobacillus*, *Bifidobacterium*

## Abstract

**Background/Objectives**: *Streptococcus mutans* (*S. mutans*) is a primary cariogenic bacterium contributing to biofilm acidogenicity and enamel demineralization. Conventional caries prevention relies mainly on mechanical plaque control and dietary modification, but probiotics have emerged as potential adjuncts for oral microbiota modulation. This pilot randomized controlled trial aimed to evaluate the short-term effects of a multi-strain probiotic containing *Lactobacillus* and *Bifidobacterium* on clinical and microbiological parameters associated with dental caries in children. **Methods**: A double-blind, randomized, placebo-controlled clinical trial was conducted in 40 children aged 6–14 years from a community setting. Participants were randomly allocated (1:1) to receive either probiotic or placebo lozenges for 30 days. Clinical assessments included the Gingival Index (GI), Plaque Index (PI), International Caries Detection and Assessment System (ICDAS), salivary pH, buffering capacity, and salivary *S. mutans* concentration. The study was preregistered (10.17605/OSF.IO/GKVUW) and ethically approved. **Results**: The intervention was well tolerated, with no adverse events reported and high participant acceptability. Despite there being no statistically significant differences in any clinical or microbiological parameter (*p* > 0.05), we found trends toward higher salivary pH, improved buffering capacity, and reduced *S. mutans* counts in the probiotic group. **Conclusions**: Short-term probiotic supplementation was safe and well accepted among children but did not produce statistically significant improvements in oral health parameters over 30 days. These findings highlight the feasibility of probiotic use in pediatric populations and support the need for larger, longer-term trials to clarify their potential role as adjuncts in caries prevention.

## 1. Introduction

Oral diseases affect an estimated 3.5 billion people worldwide, representing approximately 43% of the global population [1]. Among these, untreated dental caries in permanent teeth remains the most prevalent chronic condition globally, affecting individuals of all ages [2,3]. It is estimated that 2.3 billion people suffer from untreated caries, and more than 530 million children are affected in their primary dentition [4].

Dental caries is a multifactorial, chronic, and dynamic disease characterized by localized destruction of dental hard tissues caused by acids derived from bacterial fermentation of dietary carbohydrates [2]. The process begins with bacterial colonization and biofilm formation on tooth surfaces. These biofilms metabolize dietary sugars into organic acids, leading to a drop in intraoral pH and initiating repeated cycles of demineralization and remineralization of the enamel [5,6].

As biofilms mature, these cycles become more pronounced and, if left uncontrolled, may progress to cavitation, pain, infection, and eventual tooth loss [5,6,7,8]. The oral cavity hosts over 1500 microbial species organized in structured biofilms through adhesion processes, extracellular polymeric substances, and quorum-sensing mechanisms [9,10,11,12]. Among these microorganisms, *Streptococcus mutans* (*S. mutans*) is considered the principal cariogenic species due to its strong adherence ability, pronounced acidogenic and aciduric properties, and abundant production of extracellular polysaccharides, which foster a dysbiotic and cariogenic biofilm environment [9,13,14,15].

Given the central role of the oral microbiota in the etiology of caries and the limitations of conventional preventive measures, increasing attention has been directed toward adjunctive strategies such as probiotics [16]. According to the World Health Organization, probiotics are defined as “live microorganisms which, when administered in adequate amounts, confer a health benefit on the host” [17]. In dentistry, probiotics are being explored for their capacity to inhibit and displace pathogenic species within the oral biofilm [18,19]. Their effects may occur through several complementary mechanisms, including integration into existing biofilms, production of bacteriocins with selective antimicrobial activity, and modulation of host immune responses [8,20].

Clinical evidence supports the potential caries-preventive efficacy of certain *Lactobacillus* strains—particularly *L. reuteri*, *L. rhamnosus*, and *L. salivarius*—through inhibition of *S. mutans* and enhancement of salivary buffering capacity, but for *Bifidobacterium* strains remain limited and warrant further investigation [17,21,22,23].

Despite growing evidence of individual strain efficacy, little is known about the synergistic effects of combining *Lactobacillus* and *Bifidobacterium* species, particularly in pediatric populations. Therefore, this study aimed to evaluate the short-term effects of a multi-strain probiotic supplement containing *Lactobacillus* and *Bifidobacterium* on clinical and microbiological parameters associated with dental caries in children.

## 2. Materials and Methods

This manuscript was prepared in accordance with the CONSORT 2025 Statement [24] and the accompanying CONSORT Reporting Checklist included as Appendix A.

### 2.1. Study Registration, Ethical Approval, and Design

This randomized, double-blind, placebo-controlled, parallel-group clinical trial was conducted in accordance with the Declaration of Helsinki. The study protocol was prospectively registered on the Open Science Framework (OSF; DOI: https://osf.io/stw8q/overview) before participant enrolment. The protocol and statistical analysis plan are openly accessible in the registry record. Ethical approval was obtained from the Ethics Committee of Egas Moniz School of Health and Science (PT-566/24), following prior review by the institution’s Scientific Committee.

The trial was designed as an exploratory pilot study with a 1:1 allocation ratio and an intervention period of 30 days. No interim analyses or stopping criteria were defined. No substantive changes were made to the protocol, outcomes, or statistical plan after trial commencement.

### 2.2. Setting and Eligibility Criteria

The study was conducted between 24 April and 2 June 2025 at a community center for children in Almada, Portugal. Children aged 6–14 years who attended the center and whose parents or legal guardians provided written informed consent were eligible. Inclusion criteria were (1) age between 6 and 14 years; (2) attendance at the participating community center; and (3) provision of written informed consent by parents or guardians. Children were excluded if the following criteria were present: (1) they had a known allergy to any component of the intervention; (2) use of probiotics or antibiotics within 30 days before enrolment; (3) use of systemic antibiotics, corticosteroids, or antiseptic mouthrinses within the previous four weeks; (4) systemic diseases; or (5) absence of informed consent.

No children or guardians were involved in the trial’s design, conduct, or dissemination. As this was an exploratory pilot, no formal sample size calculation was performed. The total of 40 participants reflected all eligible children available during the recruitment period.

### 2.3. Randomisation and Blinding

The random allocation sequence was computer-generated by an independent researcher not involved in participant enrolment or assessment, using simple randomization (1:1). Allocation concealment was ensured through sequentially numbered, opaque, sealed envelopes, opened only after baseline assessment.

Participants, caregivers, investigators, and outcome assessors were blinded to group allocation. The probiotic and placebo lozenges were identical in appearance, taste, and packaging, ensuring indistinguishable administration. Unblinding was permissible only in case of a serious adverse event, which did not occur.

### 2.4. Intervention and Comparator

Participants in the experimental group (*n* = 20) received orally dissolvable probiotic lozenges containing a total of 2 × 10^9^ CFU of four microencapsulated strains: *Lactobacillus reuteri* LR92, *L. salivarius* SP2, *L. rhamnosus* GG, and *Bifidobacterium animalis* subsp. *lactis* BS01. Each lozenge also contained 80 mg of prebiotic fibres (50 mg fructooligosaccharides, 30 mg inulin), calcium, vitamin C, and vitamin D_3_ (each 15% of daily recommended intake). The bacterial concentrations of the product were ensured by the manufacturer, as they were provided directly. The received batch was stored under controlled ambient conditions as recommended by the manufacturer.

The placebo group (*n* = 20) received identical strawberry-flavoured lozenges containing no microorganisms. Both products were free of sugar, artificial colourants, preservatives, gluten, and lactose.

Both the test and controls were provided by an external collaborator. Due to confidentiality agreements, the commercial designation of the product cannot be disclosed.

Administration took place daily at the community center after the afternoon snack and supervised toothbrushing with fluoridated toothpaste (1400–1450 ppm F). Staff recorded adherence, and standardised oral hygiene reinforcement was provided weekly. The intervention lasted 30 days.

### 2.5. Outcomes and Assessments

Assessments were conducted at baseline (T_0_) and post-intervention (T_1_). Primary outcomes were salivary *S. mutans* counts and salivary pH/buffering capacity. Secondary outcomes included Gingival Index (GI) [25], Plaque Index (PI), and caries status via the International Caries Detection and Assessment System (ICDAS) [26].

Clinical examinations were performed by two calibrated dental researchers under supervision of licensed dentists. Calibration yielded 100% inter-examiner agreement for GI and ICDAS, and an ICC = 0.839 for PI.

Saliva samples were collected under stimulation, transported to the Egas Moniz laboratory, and processed within two hours. pH was measured using a calibrated potentiometer, and buffering capacity was determined by titration with 0.1 mol/L HCl in 20 µL increments up to 320 µL.

Microbiological analysis was performed under a laminar flow cabinet. Serial dilutions were plated on Mitis–Salivarius agar for *Streptococcaceae* and on blood agar for total anaerobes. Plates were incubated at 37 °C for 48 h. Colony-forming units (CFU/mL) were calculated as log_10_ CFU/mL. *S. mutans* quantification was the main microbial endpoint.

### 2.6. In Vitro Assays for Confirmation of Antimicrobial Activity

To complement in vivo findings, in vitro assays evaluated the probiotic’s inhibitory potential against *S. mutans* and other *Streptococcus* spp. Dissolved lozenges were applied to sterile discs placed on Mueller–Hinton agar inoculated with *S. mutans* ATCC 25176 and other reference strains (*S. sanguinis*, *S. anginosus*, *S. mitis*, *S. agalactiae*, *S. pneumoniae*, *S. pyogenes*, *S. bovis*, *S. salivarius*). Inhibition zones were measured after 48 h at 37 °C. In vitro sensitivity testing revealed inhibition halos around discs containing the probiotic formulation, indicating a predominantly bacteriostatic effect against *S. mutans*. Slight inhibition was also observed for *S. sobrinus*, whereas no significant effects were detected against other oral streptococci species.

### 2.7. Monitoring of Harms

Potential harms, including gastrointestinal discomfort or allergic reactions, were systematically monitored at each visit and could be reported spontaneously by staff or guardians. No adverse events were reported throughout the study.

### 2.8. Statistical Analysis

Data were analyzed using IBM SPSS^®^ Statistics v30 (IBM Corp., Armonk, NY, USA). Descriptive statistics (mean ± SD) were computed for all variables. Data normality was verified by the Shapiro–Wilk test. Depending on distribution, paired *t*-tests or Wilcoxon tests were used for within-group comparisons, and independent *t*-tests or Mann–Whitney tests for between-group comparisons. The significance threshold was set at *p* < 0.05. All analyses followed the intention-to-treat principle. Missing data were handled using pairwise deletion, as no missing outcome data exceeded 5%.

A post-hoc power analysis was conducted to determine whether the study had sufficient power to detect a clinically relevant 1−log_10_ reduction in *Streptococcus mutans* (CFU/mL).

## 3. Results

### 3.1. Participant Flow and Recruitment

A total of 40 children were assessed for eligibility and all met inclusion criteria, providing written informed consent via their parents or guardians. All participants received the assigned intervention and completed the 30-day follow-up, with no losses to follow-up or protocol deviations (Figure 1). Recruitment occurred between 24 April and 2 June 2025, and follow-up concluded on 2 June 2025.

A total of 44 children were screened for eligibility; 4 did not meet inclusion criteria and were excluded. Forty participants were randomized equally into the probiotic group (*n* = 20) and the placebo group (*n* = 20). All randomized participants received the allocated intervention, completed the 30-day follow-up, and were included in the final intention-to-treat analysis. No participants were excluded from the final analysis, and all data were analyzed according to the intention-to-treat principle.

### 3.2. Baseline Characteristics

The baseline demographic and oral health characteristics of participants are summarized in Table 1. The mean age was 10.25 ± 2.48 years (range: 6–14), with 57.5% females and 42.5% males. Most participants were of African ethnicity (77.5%), and groups were comparable across sociodemographic and behavioural variables. Most children reported regular toothbrushing (≥2 times/day) and use of fluoride toothpaste (92.5%). No statistically significant baseline differences were found between the probiotic and placebo groups.

### 3.3. Clinical and Biochemical Outcomes

#### 3.3.1. Caries Status

At baseline (T_0_), the prevalence and severity of caries lesions (ICDAS) were similar in both groups as expected. After the 30-day intervention (T_1_), no statistically significant differences were detected either within or between groups.

#### 3.3.2. Salivary pH and Buffering Capacity

Mean salivary pH increased slightly in the probiotic group after intervention (Δ = +0.30 ± 0.78), whereas values remained nearly unchanged in the placebo group (Δ = +0.11 ± 1.03). Buffering capacity showed a non-significant trend towards improvement in the probiotic group (*p* = 0.063), while it slightly declined in the placebo group (*p* = 0.733). No between-group differences reached statistical significance (*p* > 0.05; Table 2).

#### 3.3.3. Plaque and Gingival Indices

Both groups exhibited a significant reduction in plaque index (PI) between T_0_ and T_1_ (Probiotic: *p* = 0.047; Placebo: *p* = 0.045), likely reflecting reinforcement of oral hygiene. Gingival index (GI) values remained stable across time points in both groups. No statistically significant differences were observed between groups for either PI or GI (Table 3).

#### 3.3.4. Microbiological Outcomes

Quantification of salivary *Streptococcus mutans* and other bacterial groups revealed modest reductions in colony counts after intervention in both groups. In the probiotic group, mean *S. mutans* counts decreased by −0.18 log_10_ CFU/mL, while total anaerobes and α-haemolytic streptococci decreased by −0.11 and −0.10 log_10_ CFU/mL, respectively. Similar results were observed in the placebo group. However, none of these changes reached statistical significance (*p* > 0.05), and no intergroup differences were detected (Table 4).

For the probiotic group (*n* = 20), the estimated SD of paired Δ log10 values was approximately 0.83, yielding >99% power to detect a 1-log reduction in bacterial counts. Power calculations were performed in accordance with standard approaches for paired-sample power estimation based on the noncentral t-distribution.

### 3.4. Acceptability and Safety

All participants completed the intervention without adverse events. The probiotic lozenges were well tolerated and rated as pleasant in taste. Most children expressed willingness to continue use, and subjective improvements in oral cleanliness and breath freshness were more frequently reported in the probiotic group than in the placebo group. No harms or unintended effects were observed throughout the trial period.

## 4. Discussion

This randomized, double-blind, placebo-controlled clinical trial evaluated the effects of a probiotic formulation containing *Lactobacillus reuteri*, *Lactobacillus rhamnosus*, *Lactobacillus salivarius*, and *Bifidobacterium animalis* subsp. *lactis* on clinical and microbiological parameters associated with dental caries in children from a socioeconomically vulnerable population. Although no statistically significant intergroup differences were observed, the findings provide valuable insight into the feasibility, safety, and potential of probiotic interventions for oral health promotion in pediatric populations up to 30 days.

### 4.1. Clinical and Microbiological Outcomes

Across the 30-day intervention, salivary pH remained stable in both groups, with a non-significant trend toward higher values in the probiotic group, particularly at higher titration volumes (80–120 µL). This pattern is consistent with previous short-term studies reporting minimal effects of probiotics on salivary pH [27]. Similarly, buffering capacity showed a slight but non-significant improvement in the probiotic group (*p* = 0.063), contrasting with a mild decline in the placebo group. While these results suggest a potential stabilizing effect, longer interventions may be required to demonstrate significant biochemical changes, as reported by [28], who observed increased buffering capacity following nine months of probiotic-enriched milk consumption.

The plaque index (PI) decreased significantly in both groups, most likely reflecting reinforced oral hygiene behaviours and supervision during the trial rather than a specific probiotic effect. The absence of intergroup differences aligns with findings by Ebrahim et al. [29], though other studies, such as Matuq Badri et al. [30], have documented significant reductions in plaque accumulation after probiotic use. Such inconsistencies underscore the influence of strain selection, dosage, delivery vehicle, and adherence on clinical outcomes.

The gingival index (GI) exhibited a slight but non-significant reduction in both groups, corroborating reports that probiotics exert limited effects on gingival inflammation in short-term interventions [27,31]. Likewise, ICDAS assessments showed no change in lesion progression, underpinning the notion that probiotic benefits in caries prevention may require sustained use and longer observation periods.

Microbiological analyses revealed modest, non-significant reductions in *Streptococcus mutans* counts, total anaerobes, and α-haemolytic streptococci across both groups. This aligns with previous short-term studies showing either minimal or transient microbial changes [27]. In contrast, other investigations have demonstrated significant reductions in *S. mutans* levels following probiotic administration [30], suggesting that the efficacy of probiotic interventions may depend on strain specificity, CFU concentration, duration, and mode of delivery.

### 4.2. Interpretation and Influencing Factors

The absence of significant group differences can be attributed to several factors. First, the short duration (30 days) may have limited probiotic colonization and biofilm integration. Caries development and microbial modulation are chronic processes, and measurable effects often emerge only after prolonged interventions [28]. In addition, most studies have used probiotics for between 8–12 weeks or longer (Systematic Rev), and this is a comparable difference to our 4-week period. Because oral colonization of probiotic strains is typically short-lived, intervention periods below 8–12 weeks may not produce stable microbiological or clinical changes. For example, short-term trials providing probiotics for ~4 weeks reported only modest improvements and sometimes waning effects at 4-week follow-up [32,33,34]. Meanwhile, studies of 8–12 weeks show more consistent and sustained changes in clinical periodontal or gingival outcomes [35,36,37]. Our 30-day duration, justified by feasibility constraints in this pediatric sample, likely reduced the probability of detecting measurable effects. Nevertheless, our results shed light on the importance of possibly extending such intervention for longer than 4 weeks to possibly see effective results, though this should be confirmed.

Second, the delivery vehicle—a dissolvable lozenge—although acceptable and safe, may have provided limited contact time between the probiotic strains and oral surfaces, thereby reducing adherence potential [38]. Host-related factors likely also contributed. The oral microbiota of children is dynamic and influenced by diet, hygiene, and environmental exposures [39], which may increase interindividual variability and obscure probiotic effects. Additionally, the socioeconomic vulnerability of participants, often linked to suboptimal oral health behaviours and limited access to care [40,41], may have affected both baseline conditions and response to intervention.

### 4.3. Limitations and Future Perspectives

This study has limitations inherent to its pilot design. The modest sample size (*n* = 40) limited statistical power, likely contributing to the absence of significant findings. Power calculations indicate that substantially larger samples would be required to detect changes in key outcomes such as *S. mutans* counts, salivary pH, and plaque index [42]. Additionally, adherence variability and the focus on *S. mutans* alone may have underestimated the broader microbiological effects of the intervention.

Future trials should address these limitations by increasing sample size, extending intervention duration, and incorporating comprehensive microbial profiling (e.g., next-generation sequencing) to capture global biofilm shifts. Evaluating alternative delivery methods—such as dairy products, chewing gums, or toothpaste—may improve retention and colonization. Stratifying participants by socioeconomic status, caries risk, or baseline microbiota composition could help identify subgroups more responsive to probiotic interventions.

A further limitation of this study is the lack of a more detailed sociodemographic characterization of the participating children and their families. Information regarding socioeconomic status was not collected (including but not restricted to parental education level, household income category, and employment status). Likewise, data on cultural and ethnic background or language environment at home (e.g., bilingual vs. Portuguese-only households) were not available. Family structure (e.g., single-parent households or guardianship arrangements), educational support needs (such as learning difficulties or participation in social support programs), and access to community welfare services were also not assessed. The absence of these variables limits generalizability of the results across more diverse socioeconomic and cultural settings. Future studies should incorporate these dimensions to allow for better interpretation of outcomes and to identify potential equity-related differences in intervention impact.

### 4.4. Clinical Relevance

From a clinical perspective, probiotics represent a promising, non-invasive adjunct to conventional caries prevention. Their potential to modulate oral microbiota, reduce pathogenic load, and support host defenses makes them particularly attractive for high-risk populations, such as socioeconomically disadvantaged children. However, robust, well-powered, and long-term clinical trials are required to confirm their preventive efficacy, clarify optimal strain combinations and dosing regimens, and establish evidence-based recommendations for their incorporation into routine oral health programs.

## 5. Conclusions

Short-term supplementation with a multi-strain probiotic was safe, well-tolerated, and acceptable among children but did not produce significant clinical or microbiological effects over 30 days. Subtle trends toward improved salivary buffering and reduced bacterial counts suggest potential benefits that warrant confirmation in larger, longer-term trials.

## Figures and Tables

**Figure 1 nutrients-17-03604-f001:**
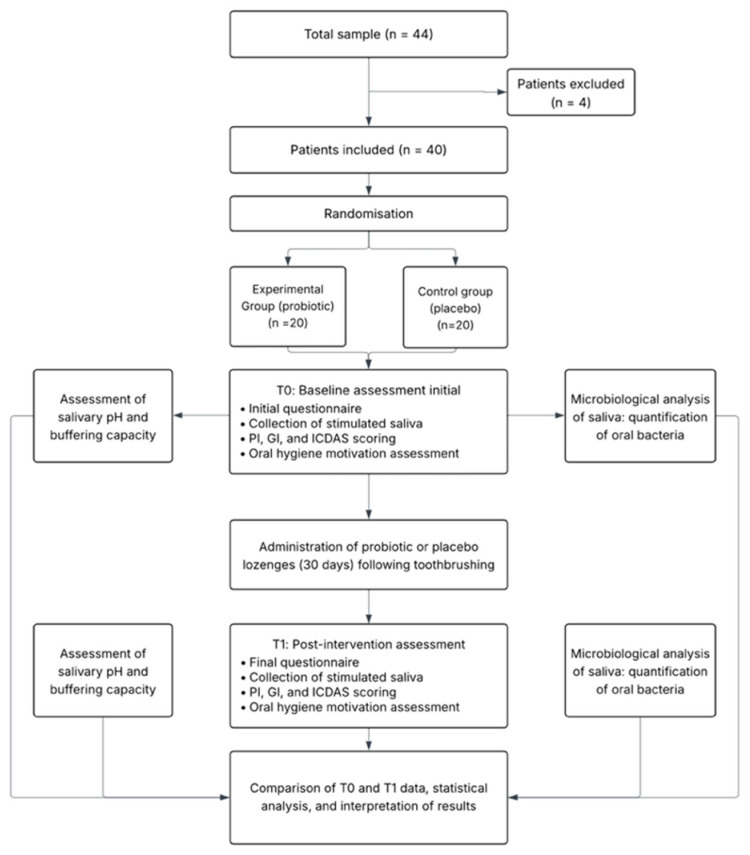
Patient flowchart. Schematic representation of the study design and timeline, showing randomisation, intervention, and outcome assessment points (T_0_ and T_1_).

**Table 1 nutrients-17-03604-t001:** Baseline characteristics of the study population.

Characteristics	Total (*n* = 40)	Control (*n* = 20)	Test (*n* = 20)
Age (years), mean ± SD (range)	10.3 ± 2.5 (6–14)	10.1 ± 2.6	10.4 ± 2.4
Sex, *n* (%)			
Female	23 (57.5%)	10 (50%)	13 (65%)
Male	17 (42.5%)	10 (50%)	7 (35%)
Ethnicity, *n* (%)			
African	31 (77.5%)	16 (80%)	15 (75%)
Caucasian/Other	9 (22.5%)	4 (20%)	5 (25%)
Toothbrushing ≥ 2/day, *n* (%)	20 (50%)	12 (60%)	8 (40%)
Use of fluoride toothpaste, *n* (%)	37 (92.5%)	17 (85%)	20 (100%)
Dental visit in past year, *n* (%)	8 (20%)	2 (10%)	6 (30%)

**Table 2 nutrients-17-03604-t002:** Salivary pH and buffering capacity at T_0_ and T_1_.

Parameter	Group	T_0_	T_1_	Δ (T_1_ − T_0_)	*p*-Value (Within-Group)	*p*-Value (Between-Groups)
Salivary pH	Probiotic	5.25 ± 0.88	5.55 ± 0.79	+0.30 ± 0.78	0.107	0.509
	Placebo	5.49 ± 0.96	5.59 ± 0.96	+0.11 ± 1.03	0.695	
Buffering capacity	Probiotic	18.79 ± 10.17	23.20 ± 11.92	+4.41 ± 9.99	0.063	0.125
	Placebo	21.01 ± 10.96	20.13 ± 8.88	−0.87 ± 0.11	0.733	

**Table 3 nutrients-17-03604-t003:** Plaque and gingival indices at T_0_ and T_1_.

Index	Group	T_0_	T_1_	Δ (T_1_ − T_0_)	*p*-Value (Within-Group)	*p*-Value (Between-Groups)
Plaque Index (PI) (%)	Probiotic	41.0 ± 26.0	27.0 ± 25.0	−14.0 ± 31.0	0.047	0.959
	Placebo	42.0 ± 33.0	27.0 ± 24.0	−14.0 ± 28.0	0.045	
Gingival Index (GI) (%)	Probiotic	8.0 ± 12.0	5.0 ± 8.0	−4.0 ± 2.0	0.207	0.354
	Placebo	4.0 ± 7.0	3.0 ± 7.0	−1.0 ± 5.0	0.346	

**Table 4 nutrients-17-03604-t004:** Streptococcaceae, anaerobes, and α-haemolytic bacteria at T_0_ and T_1_.

Parameter	Groups	T_0_ log CFU	T_1_ log CFU	Δ log CFU (T_1_ − T_0_)	*p*-Value (Within-Group)	*p*-Value (Between-Groups)
*Streptococcaceae* microorganisms	Probiotic	7.56 ± 0.57	7.38 ± 0.59	−0.18 ± 0.19	0.337	0.865
	Placebo	7.39 ± 0.45	7.17 ± 0.67	−0.22 ± 0.13	0.102	
Total anaerobic microorganisms	Probiotic	7.82 ± 0.45	7.71 ± 0.56	−0.11 ± 0.17	0.558	0.694
	Placebo	7.75 ± 0.35	7.73 ± 0.43	−0.02 ± 0.12	0.872	
Alpha-haemolytic microorganisms	Probiotic	7.24 ± 0.41	7.14 ± 0.65	−0.10 ± 0.19	0.496	0.869
	Placebo	7.27 ± 0.28	7.12 ± 0.62	−0.15 ± 0.13	0.278	

## Data Availability

Data is available as a Appendix A.

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
