# Peer review of "Effects of Probiotic Short-Term Regiment on Oral Health Parameters in Children: A Pilot Randomized Controlled Trial"

_nutrients, 2025, doi:10.3390/nu17223604_

Round 1

Reviewer 1 Report

Comments and Suggestions for Authors

This is a serious and relevant study that will be of interest to readers. It addresses an important topic related to the use of probiotics for caries prevention in socioeconomically vulnerable children. The randomized, double-blind, placebo-controlled design adds methodological strength, and the choice of probiotic strains is appropriate for the research question.

This clinical trial evaluated a probiotic formulation containing Lactobacillus reuteri, Lactobacillus rhamnosus, Lactobacillus salivarius, and Bifidobacterium animalis subsp. lactis on clinical and microbiological parameters associated with dental caries. Although no significant intergroup differences were found, the study provides valuable data on feasibility, safety, and the potential role of probiotics in oral health promotion.

The study was prospectively registered on OSF, which supports transparency, although OSF is not an official clinical trial registry recognized by ICMJE or WHO.

The limitations section is adequate in recognizing the pilot nature of the study, but the discussion is too general. The small sample size limits statistical power and likely contributed to the lack of significant findings. The authors should specify the achieved power or the required sample size to detect clinically relevant differences in S. mutans counts, salivary pH, or plaque index.

The 30-day intervention period is too short to expect a stable probiotic effect on the oral microbiota. Most studies showing measurable clinical or microbiological impact use interventions lasting 8–12 weeks or longer. This limitation should be clearly acknowledged, as it probably influenced the inconclusive results.

Author Response

We thank Reviewer 1 for the constructive and positive feedback regarding the relevance, methodological rigor, and contribution of our study. Below, we address each comment point-by-point.

Comment 1:

This is a serious and relevant study that will be of interest to readers. It addresses an important topic related to the use of probiotics for caries prevention in socioeconomically vulnerable children. The randomized, double-blind, placebo-controlled design adds methodological strength, and the choice of probiotic strains is appropriate for the research question.

This clinical trial evaluated a probiotic formulation containing Lactobacillus reuteri, Lactobacillus rhamnosus, Lactobacillus salivarius, and Bifidobacterium animalis subsp. lactis on clinical and microbiological parameters associated with dental caries. Although no significant intergroup differences were found, the study provides valuable data on feasibility, safety, and the potential role of probiotics in oral health promotion. 

The study was prospectively registered on OSF, which supports transparency, although OSF is not an official clinical trial registry recognized by ICMJE or WHO.

Our response: We appreciate the reviewer’s acknowledgement of the study’s strengths and relevance. Thank you.

Comment 2:

The authors should specify the achieved power or required sample size to detect clinically relevant differences.

Response: We have followed this remark accordingly. In the Statistical section of the Methods we added: “A post-hoc power analysis was conducted to determine whether the study had sufficient power to detect a clinically relevant 1-log10 reduction in Streptococcus mutans (CFU/mL).”. We then confirmed in the results the following: “For the probiotic group (n = 20), the estimated SD of paired Δ log10 values was approximately 0.83, yielding >99% power to detect a 1-log reduction in bacterial counts. Power calculations were performed in accordance with standard approaches for paired-sample power estimation based on the noncentral t-distribution.”

Comment 3:

The 30-day intervention period is too short to expect a stable probiotic effect. Most studies use 8–12 weeks or longer.

Response: We agree. Based on the reviewer’s guidance, we have revised the Discussion and Limitations sections to explicitly acknowledge that. The added new text now reads as follows: “

In addition, most studies have used probiotics between 8-12 weeks or longer (Systematic Rev), and this is a comparable difference to our 4-week period. Because oral colonization of probiotic strains is typically short-lived, intervention periods below 8–12 weeks may not produce stable microbiological or clinical changes. For example, short-term trials providing probiotics for ~4 weeks reported only modest improvements and sometimes waning effects at 4-week follow-up [32–34]. Meanwhile, studies of 8–12 weeks show more consistent and sustained changes in clinical periodontal or gingival outcomes [35–37]. Our 30-day duration, justified by feasibility constraints in this pediatric sample, likely reduced the probability of detecting measurable effects. Nevertheless, our results shed light on the importance of possibly extending such intervention longer than 4 weeks to possibly see effective results, though this should be confirmed.

Comment 4:

The discussion is too general. The small sample size likely contributed to the lack of significant findings.

Our response: Revised. The Discussion now more directly interprets the results in light of sample size, variability of outcomes, and known heterogeneity in pediatric oral microbiota.

Reviewer 2 Report

Comments and Suggestions for Authors

1, While the authors state that the study population consisted of children attending a community center in Almada, Portugal, the description of participant characteristics is insufficient. In particular, the socio-demographic attributes of the children remain unclear. The current Table 1 provides only limited information, making it difficult for readers to understand the context of this specific population.

To improve the clarity and generalizability of the findings, the authors should provide further details on the characteristics of the participating children and their families.

  • Socioeconomic status indicators (e.g., parental education level, household income brackets, employment status)

  • Cultural and ethnic background, including the proportion of children from immigrant or foreign-born families

  • Language environment at home (e.g., bilingual or Portuguese-only households)

  • Household structure, such as single-parent families or guardianship status

  • Educational support needs, such as learning difficulties or enrollment in social support programs

  • Access to community welfare services, if relevant to the intervention

2, The authors report using a 30-day intervention period with probiotics to effect change in oral conditions. However, many readers will reasonably question whether a 30-day period is sufficient to produce clinically meaningful changes in oral health (eg, plaque levels, gingival inflammation, microbial composition) given the chronic and gradually evolving nature of oral disease. To strengthen the manuscript, the authors should provide justification and contextualisation for selecting this relatively short timeframe, and cite previous literature showing that short-term probiotic interventions can yield measurable effects in the oral environment.

Author Response

We thank Reviewer 2 for the constructive and positive feedback regarding the relevance, methodological rigor, and contribution of our study. Below, we address each comment point-by-point.

Comment 1

1, While the authors state that the study population consisted of children attending a community center in Almada, Portugal, the description of participant characteristics is insufficient. In particular, the socio-demographic attributes of the children remain unclear. The current Table 1 provides only limited information, making it difficult for readers to understand the context of this specific population.

To improve the clarity and generalizability of the findings, the authors should provide further details on the characteristics of the participating children and their families.

Socioeconomic status indicators (e.g., parental education level, household income brackets, employment status)

Cultural and ethnic background, including the proportion of children from immigrant or foreign-born families

Language environment at home (e.g., bilingual or Portuguese-only households)

Household structure, such as single-parent families or guardianship status

Educational support needs, such as learning difficulties or enrollment in social support programs

Access to community welfare services, if relevant to the intervention.

Our response: Thank you for this insightful comment. Unfortunately, at the time of research preparation, such data was not possible to collect considering ethical concerns regarding the shortness and clearness of the questionnaire. Nevertheless, we added such a point to the discussion as a limitation that such context of the specific population should be explored in the future. This new added text reads as follows: “A further limitation of this study is the lack of a more detailed sociodemographic characterization of the participating children and their families. Information regarding socioeconomic status was not collected (including but not restricted to parental education level, household income category, and employment status). Likewise, data on cultural and ethnic background or language environment at home (e.g., bilingual vs. Portuguese-only households) were not available. Family structure (e.g., single-parent households or guardianship arrangements), educational support needs (such as learning difficulties or participation in social support programs), and access to community welfare services were also not assessed. The absence of these variables limits generalizability of the results across more diverse socioeconomic and cultural settings. Future studies should incorporate these dimensions to allow for better interpretation of outcomes and to identify potential equity-related differences in intervention impact.”

Comment 2

Justify the 30-day intervention period and cite evidence that short-term probiotic interventions can yield measurable effects.

Our response: We agree. Based on the reviewer’s guidance, we have revised the Discussion and Limitations sections to explicitly acknowledge that. The added new text now reads as follows: “In addition, most studies have used probiotics between 8-12 weeks or longer [29], and this is a comparable difference to our 4-week period. Because oral colonization of probiotic strains is typically short-lived, intervention periods below 8–12 weeks may not produce stable microbiological or clinical changes. For example, short-term trials providing probiotics for ~4 weeks reported only modest improvements and sometimes waning effects at 4-week follow-up [30-31]. Meanwhile, studies of 8–12 weeks show more consistent and sustained changes in clinical periodontal or gingival outcomes [29, 32]. Our 30-day duration, justified by feasibility constraints in this pediatric sample, likely reduced the probability of detecting measurable effects. Nevertheless, our results shed light on the importance of possibly extending such intervention longer than 4 weeks to possibly see effective results, though this should be confirmed.”

Reviewer 3 Report

Comments and Suggestions for Authors

The manuscript nutrients-3953007 has an interesting aim: to evaluate the short-term effects of a multistrain probiotic supplement containing Lactobacillus sp. and Bifidobacterium sp. on clinical and microbiological parameters associated with dental caries in children.

The study involved 40 children (6-14 years old), divided into 2 equal groups, who received probiotics or a placebo for 30 days. The study design, inclusion/exclusion criteria, and the protocol are clearly presented. The pharmaceutical form administered is orally dissolvable lozenges.

Table 1. Why do both last columns (3 and 4) have the same name (test n=20)? 

Table 2. Please verify the first column: it does not reveal the measured parameters mentioned in the Table 2 caption.

Please replace the term "index" in the header of column 1 with "parameter" (this applies to Tables 2 and 4).

Tables 2, 3, 4 = It is not necessary to repeat mean +/- SD twice in the headers, because it is mentioned in line 162.

Table 3. In the table footer, please provide the range and a brief interpretation of both indexes, PI and GI.

Is the log UFC from Table 4 the same as  log₁₀ CFU/mL from the text (lines 215 and 216)? Please use a constant notation.

The results are not significantly different, and the authors have attempted to explain them and to outline the present study's limitations.

The reviewer suggests correlating the baseline data in Table 1 with the measured parameters in Tables 2-4 to assess their influence and discuss the statistical results obtained.

Author Response

We thank Reviewer 3 for the constructive and positive feedback regarding the relevance, methodological rigor, and contribution of our study. Below, we address each comment point-by-point.

Table 1. Why do both last columns (3 and 4) have the same name (test n=20)?

Our response: We have corrected this typo.

Table 2. Please verify the first column: it does not reveal the measured parameters mentioned in the Table 2 caption.

Our response: We have corrected this typo.

Please replace the term "index" in the header of column 1 with "parameter" (this applies to Tables 2 and 4).

Our response: We have replaced them accordingly.

Tables 2, 3, 4 = It is not necessary to repeat mean +/- SD twice in the headers, because it is mentioned in line 162.

Our response: We have removed mean ± SD in the headers accordingly.

Table 3. In the table footer, please provide the range and a brief interpretation of both indexes, PI and GI.

Our response: indeed we have reported as decimals when in fact should be reported in percentage. The revised table now presents the results in percentage.

Is the log UFC from Table 4 the same as  log₁₀ CFU/mL from the text (lines 215 and 216)? Please use a constant notation.

Our response: We have corrected this typo.

The reviewer suggests correlating the baseline data in Table 1 with the measured parameters in Tables 2-4 to assess their influence and discuss the statistical results obtained.

Our response: considering the limited sample size, we limited the statistical analysis and avoided correlations. This was decided to avoid misinterpretations of correlation tests with possibly low statistical strength.

Round 2

Reviewer 1 Report

Comments and Suggestions for Authors

The work is interesting and the responses are appropriate

However, it has not yet been registered. Although it has a DOI, no visible content is available, and it has not been possible to verify its official registration.

Author Response

Dear reviewer

Thank you for you feedback. Regarding the link, we apologize for this typo. We have now revised the manuscript and have corrected the link to the following: https://osf.io/stw8q/overview. This change was highlighted in tracked changes in the revised manuscript submitted.

Reviewer 2 Report

Comments and Suggestions for Authors

Good revised.

Author Response

Thank you so much for your response and feedback. We appreciate the contributions you made to our manuscript